# Role of Genomic and Molecular Biology in the Modulation of the Treatment of Endometrial Cancer: Narrative Review and Perspectives

**DOI:** 10.3390/healthcare11040571

**Published:** 2023-02-15

**Authors:** Ilaria Cuccu, Ottavia D’Oria, Ludovica Sgamba, Emanuele De Angelis, Tullio Golia D’Augè, Camilla Turetta, Camilla Di Dio, Maria Scudo, Giorgio Bogani, Violante Di Donato, Innocenza Palaia, Giorgia Perniola, Federica Tomao, Ludovico Muzii, Andrea Giannini

**Affiliations:** 1Department of Gynecological, Obstetrical and Urological Sciences, Sapienza University of Rome, 00161 Rome, Italy; 2Department of Medical and Surgical Sciences and Translational Medicine, Translational Medicine and Oncology, Sapienza University, 00161 Rome, Italy; 3Department of Gynecologic Oncology, Fondazione IRCCS Istituto Nazionale dei Tumori di Milano, 20133 Milan, Italy

**Keywords:** endometrial cancer, molecular, genetics, target therapy, personalized treatment

## Abstract

Endometrial cancer (EC) is one of the most common gynecological malignancies in Western countries. Traditionally, loco-reginal dissemination and histological characteristics are the main prognostic factors. Nowadays, molecular and genomic profiling showed exciting results in terms of prognostication. According to the data provided by The Cancer Genome Atlas and other studies, molecular and genomic profiling might be useful in identifying patients al low, intermediate, and high risk of recurrence. However, data regarding the therapeutic value are scant. Several prospective studies are ongoing to identify the most appropriate adjuvant strategy in EC patients, especially for those with positive nodes and low volume disease. The molecular classification has offered the possibility to improve the risk stratification and management of EC. The aim of this review is to focus on the evolution of molecular classification in EC and its impact on the research approach and on clinical management. Molecular and genomic profiling might be useful to tailor the most appropriate adjuvant strategies in apparent early-stage EC.

## 1. Introduction

In developed countries, endometrial cancer (EC) is the most common gynecological malignancy [1,2] and one of the leading causes of cancer mortality among women in developed countries [3]. In 2020, more than 417,000 new EC cases were estimated to be diagnosed worldwide [4]. Its incidence has also risen due to the increase in risk factors in the female population, especially obesity and aging [4].

Currently, the gold standard of therapy includes extrafascial hysterectomy with bilateral salpingo-oophorectomy (BSO) with eventual adjuvant therapy decided based on the class of the risk of developing recurrences (chemotherapy and/or radiotherapy and/or brachytherapy). Moreover, in recent years, there has been widespread agreement on the evaluation of lymph node status through the removal of the first lymph node that drains the tumor (sentinel lymph node, LS) [5,6].

Molecular classification and traditional clinicopathological prognostic factors represent the mainstay of risk classification and are correlated with prognosis, clinical management, and the personalization of patient therapy [7,8,9].

Therapies that target molecules responsible for carcinogenesis have been elaborated for several decades, and to this day, the use of treatments that focus on molecular aberrations of the malignant tumors is considered one of the best treatment options for promising outcomes [7,8,9]. Recent preclinical studies concentrating on the disease biology have released satisfactory results, leading to the start of clinical trials to test the potential of new biological agents in EC treatment [10]. This review seeks to focus on the current classification of EC, the development of molecular classifications, and their impact on both medical research and clinical management.

We provide a critical assessment of the impact of molecular/genomic profiling in EC, focusing on current implications and further perspectives.

## 2. Materials and Methods

In November 2022, an extensive literature search was conducted by different authors to identify relevant trials on various databases (MEDLINE, Embase, Pubmed, and Cochrane). We selected all articles including the following key criteria: “carcinoma”, “endometrial cancer”, “molecular”, “genetics”, “target therapy”, and “personalized treatment”. No filter on the year of publication was set. The selected articles were rigorously reviewed and evaluated to identify studies that potentially meet the aims of this review.

Key criteria for inclusion were: (1) articles in English, (2) original studies that dealt with the molecular classification of endometrial cancer, (3) studies evaluating EC management based on molecular classification, (4) studies analyzing the prognostic role of new molecular markers, and (5) studies comparing different adjuvant therapy in four molecular subclasses.

Letters, editorials, and case reports were excluded from this review. The studies that met the inclusion and exclusion criteria were further analyzed, and relevant data were extracted and analyzed for each paper. Any discrepancies between the investigators were resolved through a consensus approach.

## 3. Molecular and Genomic Profiling

In 1983, Bokhman introduced the historic pathogenetic classification of endometrial cancer that divided EC into type I and type II [11]. Type 1 accounted for about 70–80% of ECs, consisting of moderately or well-differentiated endometrioid tumors, with positive hormone receptors and tumors more common in obese women. Type I had a favorable prognosis, manifesting in most cases in women with risk factors (smoking, early menarche, late menopause, nulliparity, lack of breastfeeding) with localized disease. In contrast, type 2 cancers, which accounted for 20–30%, had non-endometrioid histology, were poorly differentiated, were hormone receptor-negative, manifested in older women than type 1, and were independent of “traditional” risk factors and were associated with a higher risk of metastasis and poor prognosis [11]. For decades, EC risk stratification was limited by the assessment of histopathological features such as gradation, histotype, depth of myometrial invasion, and involvement of contiguous structures such as the cervix and annexes. Fortunately, in 2013, the Cancer Genome Atlas (TCGA) Research Network exceeded the limitations of CE classification by integrating molecular characterization.

Now, EC can be divided into four prognostically relevant groups based on the type of mutations and somatic copy-number variations, genome, and exome sequencing, and microsatellite instability (MSI) assay: [12] polymerase epsilon (POLE) ultramutated, MSI hypermutated, copy-number (CN) low, and CN high. Each group is related to specific progression-free survival and recurrence risk. The POLE ultramutated group is characterized by somatic mutations in the exonuclease domain of polymerase epsilon DNA. This subgroup comprises low-grade and high-grade EC and is commonly manifested in young women with lower body mass indexes. Regardless of tumor grade, it has an excellent prognosis with no recurrence. Despite the high tumor grade and nuclear atypia, the POLE ultramutated group shows less than 1% of mortality [4,13,14,15]. The MSI hypermutated group is caused by defects in DNA mismatch repair (MMR) systems. Microsatellite instability is present in 10–15% of colon cancers and is the main genetic alteration in Lynch syndrome. MutL protein homolog 1 (MLH1) promoter hypermethylation is responsible for silencing one of the key genes such as MLH1. This subgroup includes grades I–III EC. Compared to the POLE subgroup, the prognosis is intermediate, and lymphovascular space invaded (LVSI) is usually present. The most common mutations are Phosphatase and TENsin homolog (PTEN), phosphatidylinositol- 3-kinase catalytic subunit alpha (PIK3CA), phosphoinositide-3-kinase regulatory subunit 1 (PIK3RI), AT-Rich Interactive Domain-Containing Protein 5B (ARID5B) [4,14,16,17,18]. Copy-number (CN) low includes most endometrioid tumors of low grade. This subgroup presents Tumor Protein 53 (TP53) wild type and POLE wild type and is also called microsatellite stable. High levels of estrogen and progesterone receptors (ER/PR) are expressed. Furthermore, it has a low number of somatic alterations. Although the prognosis depends on the stage and the histomorphology of the tumor, it is mostly excellent [4,14,19,20]. Copy-number (CN) high is characterized by a high mortality rate. This subgroup has the worst prognosis. The genetic alterations are P53 abnormalities, with a high number of somatic alterations [4,14,19,21]. The most common tumors are serous and mixed carcinomas, and the majority are high-grade tumors, although also low-grade tumors are included. Fortunately, this subgroup represents 8–24% of EC. Table 1 details the principal characteristics of molecular subgroups.

While the TCGA study was innovative and extremely precise in characterizing patient with EC, the main drawbacks were complexity, cost and impracticality in clinical practice. Over the years, practical methodologies were developed to create a model called ProMisE (Proactive Molecular Risk Classifier for Endometrial Cancer) based on the Institute of Medicine (IOM) guidelines. This model, validated on a large cohort of patients, includes several steps in creating a molecular decision tree analysis.

The first step is the determination through immunohistochemistry (IHC) of the presence or absence of two mismatch repair (MMR) proteins: mutS homolog 6 (MSH6) and PMS2. If the analysis does not detect the presence of these proteins, EC is classified into a MMR-deficient (dMMR) subgroup. If MMR proteins are physiologically expressed, the sample is further analyzed, using PCR to identify POLE exonuclease domain mutation (“POLE EDM”). If these mutations are present, the analysis stops and is classified within the POLE ultramutated group.

In case of the absence of these mutations, the last step is to use IHC for the p53 status, wild type, or null/missense mutations [12,22,23].

To date, it is recommended to perform molecular analysis on all endometrial carcinomas according to the algorithm described. The decision to perform molecular testing depends on the resources and arrangements of each center’s disciplinary team [4]. The main goal has always been to develop practical and inexpensive molecular classification that could also be used on endometrial biopsies or curettages. The biological and molecular information of the tumor helps establish adequate therapeutic management, the radicality of surgery, and possible adjuvant or molecular therapy. The application of the ProMise molecular classification on diagnostic specimens has been extensively validated by numerous studies, which demonstrated a high concordance between these and final hysterectomy specimens [24]. Among them, one of the most important is an analysis of 947 early-stage CE patients from two large randomized trials (PORTEC-1 and 2) mainly at high/intermediate risk. The aim was to confirm and validate the prognostic significance of molecular classification and improve the risk assessment of this by correlating molecular subgroups, other genetic mutations and invasion of the lymphovascular space [25].

Mutations in the following genes were analyzed: B-Raf proto-oncogene (BRAF), cyclin-dependent kinase inhibitor 2A (CDKNA2), catenin beta 1 (CTNNB1), F-box and WD repeat domain containing 7 (FBXW7), fibroblast growth factor receptor 2 (FGFR2), fibroblast growth factor receptor 3 (FGFR3), forkhead box L2 (180 FOXL2), HRAS, KRAS, NRAS, PIK3CA, protein phosphatase 2 scaffold subunit alpha (PPP2R1A), PTEN; also the expression of ER, PR, β-catenin, AT-rich interaction domain 1A (ARID1a) and L1 cell adhesion molecule (L1CAM) were studied. The main differences between the four molecular subgroups are clinicopathological and molecular features that reflect different clinical outcomes. The tumor with P53 mutation had unfavorable prognosis, with more than 10% L1CAM expression, PPP2R1a, and FBXW7 mutations, and histologic grade 3, without hormone receptor expression. MSI tumors and no specific molecular profile (NSMP) group tumors have an intermediate prognosis. The first presented more frequently LVSI and ARID1a abnormal expression. The second was frequently graded 1 with CTNNB1 mutant. The last group with POLE mutations had a favorable prognosis that occurred more frequently in younger women, even if associated with grade 3 and PTEN mutations. Univariable analysis and multivariable analyses showed that p53-mutant, substantial LVSI, and more than 10% L1CAM expression were the strongest prognostic factors for increased risk of recurrence and decreased overall survival. In addition, EC patients carrying CTNNB1 exon 3 mutations had an increased risk of distant recurrence. Estrogen receptor (ER) positivity, phosphatidylInositol 3-Kinase/protein-kinase B (PI3K/AKT) mutations, progesterone receptor (PR) positivity, and L1CAM positivity are molecular characteristics found in tumors with worse prognosis, respectively, in 78%, 65%, 61%, and 28% and less frequently mutation of FBXW7 and FGFR2 genes in 9% and 7% of cases. These subgroups with precise molecular subtyping are particularly useful in G3 ECs and all high-risk ECs [26,27,28]. Bosse et al., considering a multicenter cohort of 381 patients with grade 3 EC, showed that the molecular subgroup with the best five-years OS and PFS was POLE, with 89% and 96%, respectively. While the worst five-years OS and PFS were for P53abn with 55% and 47% respectively. The remaining subgroups showed intermediate prognosis with five-years OS and five-years PFS 75% and 77% for MMRd, respectively, and 69% and 64% for NSMP, respectively. Patients with POLE mutant carcinomas had significantly better oncologic outcomes compared to patients of other subgroups, and p53abn remained prognostically unfavorable compared to other molecular alterations [29]. The prognostic accuracy is demonstrated for all high-risk ECs, from data of 423 participating EC samples in PORTEC-3 [30]: the worst five-years RFS at five years remains for patients with p53abn EC and the best is for POLE-mut EC, 48%, and 98% respectively.

High-risk endometrial cancers are a heterogeneous group of tumors, including non-endometrioid histotypes, with different molecular alterations and clinical outcomes. In addition to classifying these tumors into the four molecular subgroups, identifying other alterations in potentially target pathways, such as in PI3K-AKT or FBXW7-FGFR2 pathways, is useful, especially those with the worst prognosis such as p53-mutant or NSMP and the non-endometrid histotype. Further studies with a large cohort of patients are needed to determine whether these additional target pathways can have a validated clinical–therapeutic role and improve survival [31,32,33].

A topic still debated today is the role of breast cancer gene 1 (BRCA1) and BRCA2 mutations on the developing EC. The women who carried a pathogenic variant of BRCA1 and BRCA2 had a lifetime risk of breast and ovarian cancer of 40–80% and 11–40%, respectively [34]. Data available in the literature have analyzed the similarities between uterine cancer, mainly serous and serous ovarian cancer, and suggested that these two classes of tumors have common pathogenetic characteristics as well as hereditary causes [35]. BRCA1 and BRCA2 are both tumor-suppressor genes involved in a homologues recombination (HR) system which has a main role in DNA damage repair before cell replication; in fact, BRCA1 and BRCA2 mutations are commonly associated with Homologous Recombination Deficiency (HRD), including other genes indirectly involved in the pathway such as ARID1A, ATM, p53 and PTEN [36,37]. There are conflicting data between EC molecular alterations and HRD. Molecular analysis of 5540 EC demonstrated that HRD may be present with a frequency of 34% with ARID1A, ATM, and BRCA2 mutations detected in 27%, 4.6%, and 3.05%, respectively [38]. Few data are available to analyze outcomes of EC patients harboring BRCA mutation. A retrospective, multicenter study, including 64 EC patients, demonstrated no difference in median overall survival, progression-free survival and disease-specific survival between the carrier BRCA mutation cohort and BRCA wild-type cohort, but patients with BRCA mutation seemed to have more advanced disease at diagnosis [39].

## 4. Therapeutic Implications

Starting from this strong scientific evidence, in recent years, EC therapy is increasingly becoming personalized for the various subclasses. Indeed, as demonstrated by a phase III study, patients classified in the low-risk category (POLE-mutated) with early-stage disease (FIGO stage I–II) do not need adjuvant treatment because of the low recurrence rate recorded, while inconclusive data are available for advanced stages [25,40]; intermediate-risk patients may instead benefit from brachytherapy, but its use should be evaluated case by case; for high-intermediate risk patients, however, the type of adjuvant treatment recommended is different for patients with negative loco-regional lymph nodes (LVSI positive and stage II may benefit from EBRT, instead, high-grade and/or substantial LVSI may benefit from chemotherapy treatment) and unknown lymph node status (LVSI positivity and/or for stage II requires EBRT, for high-grade and/or substantial LVSI positivity adjuvant chemotherapy is recommended, finally, high-grade LVSI negative and stage II grade and endometrioid carcinomas may benefit from adjuvant brachytherapy alone); finally, high-risk patients instead need systemic adjuvant therapy (EBRT with concurrent and adjuvant chemotherapy) [4,41].

Although various studies are ongoing, no level A evidence supported the use of mutational and genomic profiling in selecting adjuvant treatments in patients with early-stage disease. To date, the MSI status has implications in selecting the most appropriate therapies in the metastatic setting [42,43].

Programmed Death Ligand1 (PD-L1) and Programmed Death-1 (PD-1) are two of the immune checkpoint-associated proteins, and these are targets of immunotherapy drugs used in various types of cancer. These proteins are expressed at high levels within the tumor microenvironment and help cancer cells escape immunosurveillance. Drugs against these proteins make cancer cells susceptible to immune system response [44]. A Phase II study KEYNOTE-158 investigated the efficacy of Pembrolizumab, a humanized anti-PD-1 monoclonal antibody, in patients with advanced MSI-H/dMMR tumor previously treated. In 2017, it was approved by the FDA for patients diagnosed with non-rewired or metastatic solid tumor [45].

From data of the KEYNOTE-028, Patrick et al. and O’Malley et al. confirmed these results encouraging survival outcomes [46,47]. Other single-agent immune checkpoint inhibitors were studied in advanced or recurrent EC with at least one prior line of platinum-based chemotherapy: nivolumab, avelumab, durvalumab (against PD-L1), and dostarlimab (against PD-1). Nivolumab monotherapy has an objective response rate (ORR) of 23% advanced EC in patients regardless of MSI status. Avelumab and durvalumab in monotherapy had ORRs of 26.7% and 43% in advanced EC dMMR tumors, respectively [48,49,50].

Oakin et al. reported preliminary data from the phase I GARNET trial, currently ongoin, which is investigating the efficacy of dostarlimab in both dMMR/MSI-H AND proficient/stable (MMRp/MSS) EC patients. The ORR is 43.1% with a good duration of response (DCR) and a manageable safety profile [51]. Therapy with Durvalumab alone, regardless of prior chemotherapy, demonstrated great effectiveness and good safety in dMMR EC, having OR 47.7%, although activity was limited in pMMR AEC [52]. Authorization for Lenvatinib associated with Pembrolizumab in advanced EC has recently been expedited by the FDA in EC, not MSI-H or dMMR, and that has not progressed after prior treatment [53]. Lenvatinib is a multikinase inhibitor that acts against vascular endothelial growth factor receptor 1 (VEGFR1), VEGFR2, VEGFR3, FGFR 1–4, KIT, RET, and platelet-derived growth factor receptor a (PDGFRa) that induces immune activation [54]. The synchronous effect of the two agents is an effective antitumor strategy. In 2019, a phase II study was published that showed the results of the efficacy of Lenvatinib combined with Pembrolizumab for patients with primary advanced or recurrent EC after one or two previous platinum-based chemotherapy treatments, regardless of MMR status, which was further evaluated [55]. In the last analysis, in 2020, a single-arm, trial KEYNOTE-146/Study 111 demonstrated that the overall ORR, median PFS, and median OS were 38%, 7.5 months, and 16.7 months, respectively, confirming the safety and efficacy of this treatment [56]. In the KEYNOTE-775/Study 309 trial, the evaluation of Pembrolizumab in combination with Lenvatinib compared with paclitaxel or doxorubicin chemotherapy demonstrated PFS, OS, and ORR of 6.6 months, 17.4 months, and 30.3%, respectively, which all significantly improved in the first arm [57]. To date, Pembrolizumab plus Lenvatinib is considered the standard second-line treatment for advanced/metastatic EC which progressed to platinum-based chemotherapy. In the United States, this treatment is approved only for MSS EC; while in Europe, it is approved in second line regardless of the MSI-H/MSS status.

## 5. Ongoing Trials

There is a growing unmet need to identify the most appropriate adjuvant strategy in EC patients, especially for those with positive nodes and low-volume disease [58,59]. Several prospective study are ongoing to test various adjuvant strategies in those patients [60].

Among the most important clinical trials, there is the RAINBO—umbrella program, which investigating new adjuvant therapies in EC patients. In this trial, EC patients are set in one of the four RAINBO trials depending on their cancer’s molecular profile. The p53abn-RED is an international, multicenter, phase III randomized trial that comprises the p53-mut treated with adjuvant chemoradiation and Olaparib for two years versus adjuvant chemoradiation only. The MMRd-GREEN trial is an international, multicenter, phase III randomized study for MMRd EC patients that brings into comparison adjuvant pelvic external beam radiotherapy and Durvalumab for one year with adjuvant pelvic external beam radiotherapy alone. The NSMP-ORANGE trial is for no specific molecular profile EC patients that are treated with adjuvant pelvic external beam radiotherapy followed by oral progestins (medroxyprogesterone acetate or megestrol acetate) for two years. The last is a POLEmut-BLUE trial for POLE mutant EC patients. The POLEmut-BLUE trial is an international, multicenter, single-arm, phase II trial that investigates the safety of de-escalation of adjuvant therapy. In particular, patients with stage I and II no receive adjuvant therapy and patients with stage III receive pelvic external beam radiotherapy or no receive adjuvant therapy. The aim of the whole RAINBO research is to associate data and tumor material of the four RAINBO clinical trials to make translational research and compare molecular profile-based adjuvant therapy to standard adjuvant therapy according to means of effectiveness, toxicity, quality of life, and cost-utility [61].

Additionally, PORTEC-4a is investigating various treatment modalities in stage I–II high-intermediate risk EC patients based on their molecular profile [62]. Other prospective studies investigating the adoption of different strategies in both adjuvant and metastatic setting are ongoing.

## 6. Discussion

Endometrial carcinoma has an altogether favorable prognosis. The surgical approach, when applicable, depending on the tumor extent and on the preoperative evaluation of the patient’s fragility is the mainstay of early treatment for EC [63]; on the other hand, adjuvant therapy can only be carried out in the context of an accurate personalized therapy, especially since patients affected by EC are often elderly patients with comorbidities, such as hypertension and diabetes, and for this reason, the best effort has to be made to reduce morbidity and improve results. Since the TCGA published the first results, many efforts have been made to incorporate histological evaluation in molecular tests, with the aim of achieving an even more specific staging for every single patient. This has led to a better understanding of tumor biology, also offering the possibility to improve the disease’s diagnosis and prognosis. Moreover, the use of molecular classification has brought a great advantage by allowing a precise selection of patients that benefit from systemic treatments, such as chemotherapy, radiotherapy and immunotherapy. In recent years, also, radiomic analysis has permitted an additional risk stratification in patients affected by endometrial carcinoma, offering the possibility to obtain information otherwise not visible to the human eye. Recent studies had investigated the role of radiomic analysis based on preoperative magnetic resonance imaging (MRI) for EC risk stratification. Bi Cong et al., from a cohort of 717 patients with EC, developed a radiomic model that demonstrated good performance to predict high risk with area under the curve (AUC) in the validation group of 0.845. Interestingly, the accuracy becomes almost excellent, with an AUC of 0.919, if clinical features were considered with radiomic features [64]. Similar results were confirmed in subsequent studies considering preoperative MRI or other second-line imaging investigation [65,66], but Mor et al., from a multicenter and retrospective study of 498 EC patients, obtained promising results developing and validating a radiomic model based on ultrasound images, and they considered the first-line imaging investigation for endometrial cancer and most often used in gynecology less expensive and easier to perform. In the validation test, the radiomics model had a sensitivity of 58.7% and specificity of 85.7% in differentiating high-risk EC from other cancers [67]. These data suggest that radiomic analysis could help in choosing the most appropriate surgical management even before the results of molecular analysis. Furthermore, due to the elevated costs of the genetic and molecular tumor evaluation, a hybrid approach has been introduced known as radio-genomics, which could allow on the one hand the fall of the costs of the processing and the analyzing of the histologic samples and on the other hand a faster and more reproducible analysis of the intrinsic characteristics of these complex diseases, as well as their behavior, before surgical treatment. Unfortunately, to date, the studies are limited. Radiomics models had developed to predict PD1 expression and association with Lynch Syndrome in 100 EC patients or determine DNA mismatch repair deficiency (MMR-D) in 150 patients [68,69].

Since endometrial carcinoma represents an emerging disease, also in patients in pre-menopause and where the age of the first pregnancy tends to be older, molecular analysis could also be used to choose the therapeutic strategy for the conservative treatment of lesions that anticipate EC. Zhang et al. in a retrospective analysis of 59 patients with EC and endometrial atypical hyperplasia/endometrial intraepithelial neoplasia (EAH/EIN) evaluated how molecular classification might predict response to conservative treatment and which subclasses are at the highest risk of evolution. The complete response was 100% in the POLEmut group and 71.43% in copy number-low mutation (CNL), demonstrating a good prognosis for these subgroups. However, the prognosis of copy number-high mutation (CNH) and the MSI-H group was significantly worse with 33.3% and 25% of complete response, respectively [70]. In another analysis of 89 EC patients, the aim was to assess the strength of various clinicopathological indicators for the prediction of treatment efficacy, and the results demonstrated no association between prognosis among ER, PAX2, PTEN or Ki-67 expression in the initially untreated AH or EEC groups, but expression >50% PR expression had the highest complete response in both the EEC and AH groups [71]. In a study, 117 cases, initially diagnosed as endometrial hyperplasia, were histopathologically reevaluated by the EIN diagnosis category to establish immunohistochemically the expressions of PTEN and b-catenin. From the results of this analysis, the combination of PTEN-negative/b-catenin-positive may become the reliable marker for detecting EIN, considering these markers predictive of disease progression [72]. A selection of the most recent studies available on the main scientific databases represents the principal strength of this review of the literature. This work also has various limitations, which are mostly due to the intrinsic nature of the work itself. To date, few data are available to support these results, and more studies are needed to validate this scientific evidence that could revolutionize the management of endometrial disease.

## 7. Conclusions

The molecular classification has offered the possibility to improve the risk stratification and management of EC. In recent years, various treatments, including chemotherapy, radiation therapy, immune checkpoint, and molecular targeting therapy, have been studied to obtain tailored therapy according to clinical data and molecular-genetics characteristics. High response rates are observed from the results of various trials about the efficacy of immune checkpoint inhibitors, especially in patients with dMMR. Therefore, in recent years, various agents have been studied in monotherapy and combination with chemotherapy or other molecules, confirming their good effectiveness. Perhaps in the next few years, the results of ongoing studies will define these agents as the new first-line treatment standard in advanced or recurrent EC and compare radiation therapy with radiation therapy plus checkpoint inhibition.

The p53 group has the worst prognosis among all EC subgroups, although it represents a small percentage of cases. Therefore, new therapeutic strategies have shown promising results. PARP inhibitors act on homologous recombination deficits, and specific antibodies may act in tumors with the overexpression of human epidermal growth factor receptor 2 (HER2) [73]. Ongoing studies are comparing chemoradiation with chemoradiation plus PARP inhibitors with the aim to define the efficacy of these therapeutic strategies [61,74]. The ongoing PORTEC 4a and the RAINBO umbrella program are the first prospective and randomized trials that may overcome the current limitations in the management of EC subclasses with the aim for tailored adjuvant treatment using the molecular profile and provide a step toward precision medicine in EC.

## Figures and Tables

**Table 1 healthcare-11-00571-t001:** Principal characteristics of molecular subgroups.

Type	Mutations	Common Features	Prognosis
POLE ultramutated	Somatic mutations in the exonuclease domain of polymerase epsilon DNA	Endometrioid carcinoma Young women Low-grade Stage IA LVSI negative or focal	Excellent
CN-low	Low number of somatic alterations TP53 wild type POLE wild type High-level ER/PR	Endometrioid carcinoma Low grade Stage I–III LVSI negative or focal	Good
MSI hypermutated	DNA mismatch repair systems defects	Endometrioid carcinoma Low and high grade Stage I–III Frequently LVSI	Intermediate
CN-high	High number of somatic alterations P53 abnormalities	Non-endometrioid (serous, clear cell, undifferentiated, carcinosarcoma, mixed) and endometrioid carcinoma High-grade tumors Frequently DMI and LVSI Stage I–IV	Worst prognosis

POLE: Polymerase epsilon; CN: copy-number; MSI: microsatellite instability; LVSI: lymphovascular space invaded; DMI: Deep Miometrial Infiltration; TP53: Tumor Protein 53; ER: Estrogen Receptor; PR: Progestin Receptor.

## Data Availability

All data are included in the main text.

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
