# Peer review of "Role of Genomic and Molecular Biology in the Modulation of the Treatment of Endometrial Cancer: Narrative Review and Perspectives"

_healthcare, 2023, doi:10.3390/healthcare11040571_

Round 1

Reviewer 1 Report

I read with great interest the Manuscript titled " Role of genomic and molecular biology in the modulation of the treatment of endometrial cancer: narrative review and future perspectives " which falls within the aim of the Journal.

In my opinion, the topic is interesting enough to attract the readers’ attention. Nevertheless, authors should address the following points to improve the quality of this manuscript:

- The whole text should be corrected by a native English speaker in order to make the work clearer and more readable and typos should be corrected. Example: last paragraphs better headed as “Therapeutic implications” and “Ongoing trials” (plural).

-The cited references are mostly within the last 5 years, but research-group-self-citing rate is high.

-Although it is a narrative review, material and methods and inclusion/exclusion criteria should be better clarified by extending their description.

-Authors should discuss robust pieces of evidence about the use of new strategies for endometrial cancer early diagnosis, even using artificial intelligence and novel biomarkers (authors may refer to: PMID: 34714568).

-Are there any data in the literature on molecular analysis for endometrial hyperplasia with atypia? If so, can the molecular assessment of such patients predict the development of aggressive cancer?

-Can this assessment also be applied to guide the future approach for patients who wish to undergo fertility sparing treatment?

Author Response

Point 1: The whole text should be corrected by a native English speaker in order to make the work clearer and more readable and typos should be corrected. Example: last paragraphs better headed as “Therapeutic implications” and “Ongoing trials” (plural).

Response 1: The whole text has been revised by a native English speaker and typos have been corrected as well as the title of paragraphs

Point 2: The cited references are mostly within the last 5 years, but research-group-self-citing rate is high.

Response 2: To reduce the self-citation rate of the research team, 3 refences, that include some authors included in the writing of this study (D’Oria O et al. Molecular Marker an endometrial cancer; Bogani G Management of endometrial cancer: issue and controversies; Bogani G et al, Low volume in endometrial cancer: the role of micrometastasis and isolated tumor cells), were removed.

Point 3: Although it is a narrative review, material and methods and inclusion/exclusion criteria should be better clarified by extending their description.

Response 3: Material and Methods section has been provided and inclusion/exclusion criteria have been clarified. (lines 75-89)

Point 4: Authors should discuss robust pieces of evidence about the use of new strategies for endometrial cancer early diagnosis, even using artificial intelligence and novel biomarkers (authors may refer to: PMID: 34714568).

Response 4: New strategies for endometrial cancer early diagnosis have been described, in particular radiomics and radiogenomics based on preoperative radiological examination, such as Magnetic Resonance Imaging.  PMID: 34714568 has been included in the references. (lines 337-362)

Point 5: Are there any data in the literature on molecular analysis for endometrial hyperplasia with atypia? If so, can the molecular assessment of such patients predict the development of aggressive cancer?

Response 5: The data available in the literature on the use of molecular analysis and its possible prognostic role, although few, have been reported and analyzed. (lines 363-368)

Point 6: Can this assessment also be applied to guide the future approach for patients who wish to undergo fertility sparing treatment?

Response 6: Although data are scarce and patient cohorts analyzed are limited, findings that could support use of molecular analysis to guide management of patients with EC undergo fertility sparing treatment are reported.  (lines368-382)

Reviewer 2 Report

The paper proposed by H Cuccu et al., and entitled “Role of genomic and molecular biology in the modulation of the treatment of endometrial cancer: narrative review and future perspectives” is a review of the molecular characterization of endometrial cancers currently performed in order to develop personalized therapies. The paper is well written and the bibliography adequate and well analyzed.

As main remarks:

1)    The development of Endometrial Cancers (EC) in women with germline BRCA1/2 mutations is not mentioned although this genetic alteration is part of the “genomic and molecular biology” of EC and has prognostic and therapeutic implications. This could be at least mentioned in the frame of this review, focusing on main practical aspects:  when to suspect BRCA-associated EC if the analysis has not been performed, what consequences on prognosis and therapeutic approach.

2)    The description of the different molecular types of EC is well conducted. But an additive table would largely help the reader to catch the main characteristics of these different tumor types.

3)    Could it be possible to more accurately describe the algorithm of the sequential analyses of tumor specimens (lines 113-116). For instance, it is stated (line 115) that “If the analysis is negative for the research of these proteins it is classified as dMMR subgroup. So the sample is further analyzed to identify POLE mutation”. Is the search for POLE mutation restricted to cases of the dMMR subgroup ? Please specify.

4)    It is not possible to perform an extensive molecular analysis for all cases of EC. Could you underline which analyses are to be performed systematically ?

Minor points:

1)    In the title, the word future might be discarded.

2)    Lines 165-166 “The prognostic accuracy of surrogate markers may be generalized to non-endometrioid histotypes…” Is it not the case ? The value of the markers described are not restricted to endometrioid tumors since the text mention (lines 104-105) serous carcinoma and mixed tumors (carcino-sarcoma ?) as potential phenotypes of CN high tumors. Please specify.

3)    Line 180: …” in patients with advanced EC and previously treated with MSI-H/dMMR tumors (?). Is it “in patients with advanced MSI-H/dMMR tumor previously treated ?.

Author Response

Point 1: The development of Endometrial Cancers (EC) in women with germline BRCA1/2 mutations is not mentioned although this genetic alteration is part of the “genomic and molecular biology” of EC and has prognostic and therapeutic implications. This could be at least mentioned in the frame of this review, focusing on main practical aspects:  when to suspect BRCA-associated EC if the analysis has not been performed, what consequences on prognosis and therapeutic approach.

Response 1: The authors mentioned the relevance of BRCA1-2 mutations but the risk and correct management of BRCA carrier patients is still debated. The evidence concerning the molecular alterations of both BRCA1/2 and the Homologous Recombination system DNA of damage repair have a crucial role for use and validation of new therapeutic strategies, currently analyzed in ongoing trials. (lines 210-228)

Point 2: The description of the different molecular types of EC is well conducted. But an additive table would largely help the reader to catch the main characteristics of these different tumor types.

Response 2: The table n.1 with the description of the different molecular types of EC with the principal characteristics of each has been realized

Point 3: Could it be possible to more accurately describe the algorithm of the sequential analyses of tumor specimens (lines 113-116). For instance, it is stated (line 115) that “If the analysis is negative for the research of these proteins it is classified as dMMR subgroup. So the sample is further analyzed to identify POLE mutation”. Is the search for POLE mutation restricted to cases of the dMMR subgroup ? Please specify.

Response 3: The algorithm of the sequential analyses of tumor specimens has described was described in a clearer and more detailed way. (lines 142-150)

Point 4: It is not possible to perform an extensive molecular analysis for all cases of EC. Could you underline which analyses are to be performed systematically ?

Response 4: Molecular analyses performed systematically have been reported. (lines 151-153)

Point 5: In the title, the word future might be discarded.

Response 5: The word “future” has been discarded.

Point 6: Lines 165-166 “The prognostic accuracy of surrogate markers may be generalized to non-endometrioid histotypes…” Is it not the case ? The value of the markers described are not restricted to endometrioid tumors since the text mention (lines 104-105) serous carcinoma and mixed tumors (carcino-sarcoma ?) as potential phenotypes of CN high tumors. Please specify.

Response 6: The reported concept has been clearly explained. (lines 202-209)

Point 7: Line 180: …” in patients with advanced EC and previously treated with MSI-H/dMMR tumors (?). Is it “in patients with advanced MSI-H/dMMR tumor previously treated ?.

Response 7: The grammatical form has been corrected. (line 260)

Reviewer 3 Report

The manuscript "Role of genomic and molecular biology in the modulation of the treatment of endometrial cancer: narrative review and future perspectives" is an interesting work on the potential role of genomic and molecular biology in the management of endometrial cancer. The work is useful for the readers and well-structured. The design of the project is appropriate and the results are significant. The English language is acceptable. The structure of the manuscript is described as a narrative review of literature, but the authors have to be more specific, because, considering the length of the text, it looks more like a commentary. Otherwise it is a good manuscript and it could be considered for publication.

Author Response

Point 1: The manuscript "Role of genomic and molecular biology in the modulation of the treatment of endometrial cancer: narrative review and future perspectives" is an interesting work on the potential role of genomic and molecular biology in the management of endometrial cancer. The work is useful for the readers and well-structured. The design of the project is appropriate and the results are significant. The English language is acceptable. The structure of the manuscript is described as a narrative review of literature, but the authors have to be more specific, because, considering the length of the text, it looks more like a commentary. Otherwise it is a good manuscript and it could be considered for publication

Response 1: The authors proceeded to expand the manuscript and to describe in detail the main dematics of the review.